# Hydrophilic Poly(glutamic acid)-Based Nanodrug Delivery System: Structural Influence and Antitumor Efficacy

**DOI:** 10.3390/polym14112242

**Published:** 2022-05-31

**Authors:** Yifei Guo, Yiping Shen, Bo Yu, Lijuan Ding, Zheng Meng, Xiaotong Wang, Meihua Han, Zhengqi Dong, Xiangtao Wang

**Affiliations:** 1Institute of Medicinal Plant Development, Chinese Academy of Medical Sciences & Peking Union Medical College, No. 151, Malianwa North Road, Haidian District, Beijing 100193, China; ffguo@163.com (Y.G.); martha411@163.com (Y.S.); yywyzb@163.com (B.Y.); 13651079648@163.com (L.D.); mz960428@163.com (Z.M.); ttwang0521@163.com (X.W.); hanmeihua727@163.com (M.H.); 2Key Laboratory of Bioactive Substances and Resources Utilization of Chinese Herbal Medicine, Ministry of Education, No. 151, Malianwa North Road, Haidian District, Beijing 100193, China; 3Key Laboratory of New Drug Discovery Based on Classic Chinese Medicine Prescription, Chinese Academy of Medical Sciences, No. 151, Malianwa North Road, Haidian District, Beijing 100193, China; 4Beijing Key Laboratory of Innovative Drug Discovery of Traditional Chinese Medicine (Natural Medicine) and Translational Medicine, No. 151, Malianwa North Road, Haidian District, Beijing 100193, China

**Keywords:** poly(amino acid), isoforms, doxorubicin, nanoparticle, pH-sensitive

## Abstract

Poly(amino acids) have advanced characteristics, including unique secondary structure, enzyme degradability, good biocompatibility, and stimuli responsibility, and are suitable as drug delivery nanocarriers for tumor therapy. The isoform structure of poly(amino acids) plays an important role in their antitumor efficacy and should be researched in detail. In this study, two kinds of pH-sensitive isoforms, including α-poly(glutamic acid) (α-PGA) and γ-PGA, were selected and used as nanocarriers to prepare a nanodrug delivery system. According to the preparation results, α-PGA can be used as an ideal drug carrier. Selecting doxorubicin (DOX) as the model drug, an α-PGA/DOX nanoparticle (α-PGA/DOX NPs) with a particle size of 110.4 nm was prepared, and the drug-loading content was 66.2%. α-PGA/DOX NPs presented obvious sustained and pH-dependent release characteristics. The IC_50_ value of α-PGA/DOX NPs was 1.06 ± 0.77 μg mL^−1^, decreasing by approximately 8.5 fold in vitro against 4T1 cells after incubation for 48 h. Moreover, α-PGA/DOX NPs enhanced antitumor efficacy in vivo, the tumor inhibition rate was 67.4%, increasing 1.5 fold over DOX injection. α-PGA/DOX NPs also reduced the systemic toxicity and cardiotoxicity of DOX. In sum, α-PGA is a biosafe nanodrug delivery carrier with potential clinical application prospects.

## 1. Introduction

Cancer is a significant threat to humans, and the exploration of its treatment remains ongoing [1,2,3]. Drug-loaded nanoparticles (NPs) are constructed by loading hydrophobic drugs on amphiphilic nanocarriers via molecular interactions [4,5]; the particles’ size ranges from 10 to 1000 nm. Because of their small size, drug-loaded NPs containing a large dose of therapeutic factors can be delivered and accumulate in tumor tissue effectively through the enhanced permeability and retention (EPR) effect [6,7]. Based on these advantages, drug-loaded NPs have become an ideal form of anticancer therapy [8,9,10].

As a component of the drug delivery system, nanocarriers play an important role [11,12]. As the traditional polymeric matrix, amphiphilic copolymers are appropriate for the preparation of a nanoscale drug delivery system due to the assembly behavior between nanocarriers and bioactive agents [13,14,15]. During the assembly procedure, the hydrophobic portion of amphiphilic copolymers can entrap hydrophobic agents via hydrophobic interactions to form the core of the NPs, and the hydrophilic portion acts as the shell to provide aqueous solubility [16,17]. Poly(amino acids) are natural substances with excellent biocompatibility and biodegradability [18,19,20]. Moreover, poly(amino acids) have an abundance of functional groups, including amino groups [21,22,23], carboxyl groups [24], and sulfhydryl groups [25], which can interact with drugs, targeting agents, and other active agents. Based on these advantages, amphiphilic copolymers containing a poly(amino acid) portion have attracted broad attention [26,27], and a variety of nanodrug delivery systems have been constructed [28,29].

Poly(glutamic acid) (PGA), as a typical poly(amino acid), has free carboxyl groups in its structure, which can provide a negative charge to interact with positively charged bioactive agents [30,31]. According to the structure of glutamic acid, PGA has two isoforms, α-PGA and γ-PGA [24]. γ-PGA is produced by several microorganisms of the Bacillus species [32,33], and α-PGA by chemical synthesis [34,35,36]. Due to their excellent hydrophilicity, biosafety, and biodegradability, a lot of PGA-based composites are employed for building nanoscale delivery systems [37,38], including hydrogels [39], nanofibers [40], monoliths [30], polymersomes [41], and NPs [42]. For these PGA-based nanoscale delivery systems, doxorubicin (DOX) is commonly selected as the model drug, as it can form stable aggregates via ionic interactions [43,44,45]. Although these PGA-based DOX delivery systems show some advantages, most of them still present an unsatisfactory drug-loading content (DLC).

An effective method to enhance the DLC is to simplify the composition of the nanocarriers. In our previous study, hydrophilic polymers, such as polyethylene glycol and oligoethylene glycol derivatives, were utilized as nanocarriers to prepare drug-loaded NPs [46,47]. Compared with the traditional amphiphilic copolymers, drug-loaded NPs prepared from hydrophilic polymers possess optimized DLC and therapeutic activity. Based on these results, PGA, as a hydrophilic poly(amino acid), might be able to be utilized as a nanocarrier to construct nanodrug delivery systems.

To investigate the properties of PGA as nanocarrier, α-PGA and γ-PGA were utilized to construct DOX NPs via the antisolvent precipitation method in this work. After successful preparation, the physiochemical properties, morphology, and in vitro release characteristics of the PGA/DOX NPs (α-PGA/DOX NPs) were investigated. The in vitro antitumor efficacy against the 4T1 cell line and in vivo antitumor activity in 4T1 tumor-bearing mice were also estimated.

## 2. Materials and Methods

### 2.1. Materials

α-PGA was synthesized according to a previous reference [48], γ-PGA was purchased from Sigma-Aldrich (Beijing, China). Adriamycin hydrochloride (DOX·HCl) and docetaxel (DTX) were purchased from Beijing Ouhe Technology Co., Ltd. (Beijing, China); ibuprofen (IBU), resveratrol (RES), and nifedipine (NIF) were purchased from HEOWNS Biochem Technologies LLC (Tianjin, China); oxcarbazepine (OXC) and methotrexate (MTX) were purchased from Adamas Reagent Co., Ltd. (Shanghai, China); podophyllotoxin (POD) was purchased from Macklin Biochemical Co., Ltd. (Shanghai, China); and the adriamycin hydrochloride injection was purchased from Shenzhen Main Luck Pharmaceuticals Inc., Ltd. (Shenzhen, China). The dialysis bag (MWCO 8000–14,000 Da) was purchased from Spectrum Laboratories Inc. Acetonitrile and methanol were chromatographically pure, while the other reagents and solvents were analytical grade and used without further purification.

### 2.2. Animals and Cell Line

The mouse breast cancer (4T1) cell line was purchased from Peking Union Medical College Cell Center and cultured in RPMI-1640 medium containing 10% fetal bovine serum, 100 units of penicillin, and streptomycin under the condition of 37 °C and 5% CO_2_.

BALB/c female mice (20 ± 2 g) were purchased from Beijing HFK Bioscience Co., Ltd. (Beijing, China). The mice were raised in an SPF-level laboratory animal room, kept on a 12 h light–dark cycle, and provided a standard diet of food and water ad libitum. All the experimental procedures complied with the Guidelines and Policies for Ethical and Regulatory for Animal Experiments and approved by the Animal Ethics Committee of Peking Union Medical College (Beijing, China).

### 2.3. Preparation of Drug-Loaded PGA NPs

The preparation method of drug-loaded PGA NPs was the reverse solvent precipitation and high-pressure homogenizer technique [49]. Briefly, 20 mg of DOX·HCl and triethylamine (TEA, 1.5-fold molar ratio vs. DOX·HCl), IBU, OXC, DTX, RES, POD, NIF, and MTX were dissolved in 0.5 mL of N, N-dimethylformamide (DMF) separately. Next, 20 mg of PGA was dissolved in 0.5 mL of DMF. These two DMF solutions were mixed and injected into deionized water (5 mL) at room temperature. Then, the dialysis method was utilized to remove the organic solvent and free drug. After being transferred into a dialysis bag (MWCO 8000–14,000), the mixture was dialyzed against 1 L deionized water, which was replaced by new deionized water (1 L) every 1 h for 4 h. The desired NPs were then homogenized at 1600 bar 10 times at 25 °C. After successful preparation, the volume of drug-loaded NPs was recorded.

### 2.4. Measurement of the DLC and Encapsulation Efficiency (EE)

Briefly, 2 mL of NP solution was lyophilized and weighed precisely, then 1 mL chromatographic methanol was added. After dissolving completely, the methanol solution was centrifuged at 13,000 rpm min^−1^ for 30 min and diluted with chromatographic methanol. The actual concentration of drug in the NPs was detected by HPLC (DIONEX, UltiMate3000, Sunnyvale, CA, USA) with the fluorescence/UV quantification method. The measurement was conducted according to previous reports [50,51]. The calculation formulas of DLC and EE are as follows:

DLC% = (weight of loaded drug/weight of NPs) × 100%.


EE% = (weight of loaded drug/weight of feed drug) × 100%.


### 2.5. Characterization of NPs

These drug-loaded NPs were diluted to a concentration of 1 mg mL^−1^ with deionized water, and the average hydrodynamic diameter (D*_h_*), polydispersity index (PDI), and zeta potential were measured using a Malvern Zetasizer 3000 system (Malvern Instruments Ltd., Malvern, UK). Measurements were conducted under a backscattering detection model (θ = 173°); a 4 mV He–Ne laser was utilized, and the wavelength was 633 nm. All samples were detected at 25 °C three times.

### 2.6. Morphology of α-PGA/DOX NPs

The morphology of the α-PGA/DOX NPs was observed using a scanning electron microscope (SEM, Hitachi Limited, Tokyo, Japan). The sample solution (10 μg mL^−1^) was dropped on a clean silica gel sheet to dry, fixed with conductive adhesive, and then sprayed with gold for 6 min under negative pressure and a current of 30 mA, before the voltage was raised to 30 mV.

### 2.7. Fourier-Transform Infrared (FT-IR) Measurement

FT-IR spectra of samples were analyzed with a spectrum 100 (PerkinElmer, Waltham, USA), spectra were recorded in the range of 400–4000 cm^−1^. The KBr disk method was utilized; sample powders (10 mg) and KBr powder (100 mg) were mixed to obtain KBr disks.

### 2.8. Stability Study

The stability of α-PGA/DOX NPs was evaluated according to D*_h_* and PDI. The storage stability of α-PGA/DOX NPs was measured at 4 °C. D*_h_* and PDI of these NPs were recorded at 0, 2, 4, 6, 8, 10, 15, and 30 days. For the media stability, α-PGA/DOX NPs solution was mixed with 1.8% saline solution (1/1, *v*/*v*), 10% glucose solution (1/1, *v*/*v*), 2 × phosphate buffer saline (1/1, *v*/*v*), and plasma (1/4, *v*/*v*), then the mixture was incubated at 37 °C. After being incubated for 0, 2, 4, 6, and 8 h, the particle size and PDI were detected separately. All samples were measured in triplicate.

### 2.9. Study on the Release of α-PGA/DOX NPs In Vitro

Cumulative drug release curves of NPs were plotted by the dialysis method. A total of 2 mL of α-PGA/DOX NP aqueous solution and DOX·HCl solution were transferred into dialysis bags; these dialysis bags were placed into 50 mL of PBS solution (pH = 7.4 and pH = 5.5) containing 0.5% sodium dodecyl sulfate at 37 °C, the concentration of DOX was 2 mg mL^−1^. At the set time point, 1 mL of release medium was removed; meanwhile, the same volume of fresh media was added. The concentration of DOX was analyzed by HPLC, and each sample was repeated three times to calculate the cumulative percentage of drug release and draw the cumulative release curve.

### 2.10. Antitumor Effect In Vitro

The 4T1 cells were cultured with different concentrations of α-PGA/DOX NPs, and the cytotoxicity of the NPs was investigated by the CCK-8 method according to the previous reports [52,53]. 4T1 cells in the logarithmic growth phase were seeded in a 96-well plate at a density of 8 × 10^3^ cells per well at 37 °C and 5% CO_2_. After a 24 h incubation, the DOX solution and α-PGA/DOX NPs were diluted to 0.01, 0.1, 0.5, 1, 2, 5, 10, 20, and 100 μg mL^−1^ with fresh RPMI-1640 and added into the well; the injected volume was 150 μL per well. The culture medium was replaced after incubation for 48 h before 10 µL CCK-8 solution was added to each well, then incubation was continued for 1.5 h. The optical density (OD) value was measured with a microplate reader at 450 nm wavelength to determine the cell viability. The half-maximal inhibitory concentration (IC_50_) was determined using the GraphPad Prism 5 software. The formula for calculating the cell inhibition rate is as follows:
Cell inhibition rate (%) = (1 − OD value of the sample groups/OD value of the control group) × 100%.


### 2.11. Investigation of Antitumor Efficacy

The 4T1 tumor-bearing mice model was established by subcutaneous injection of a 4T1 cell suspension (0.2 mL) with a concentration of 1 × 10^7^ mL^−1^ into the right axilla of BALB/c mice. When the tumor volume grew to about 150 mm^3^, the mice were randomly divided into four groups (*n* = 10): the negative control group, the positive control group, the blank control group, and the test group, which was treated with 5% glucose solution, DOX injection (3 mg Kg^−1^), α-PGA (1.5 mg Kg^−1^), and α-PGA/DOX NPs (3 mg Kg^−1^, DOX equivalent concentration) via tail vein injection separately. The mice were treated every two days for seven consecutive times, their tumor volumes were measured, and their body weight was taken. After being sacrificed by cervical dislocation, the tumor tissue, heart, liver, and spleen were extracted and weighed, and the tumor inhibition rate (TIR), liver index, spleen index, and heart index in vivo were calculated.

TIR (%) = (1 − mean tumor weight of treatment group/mean tumor weight of negative control group) × 100%.


Liver index (%) = (liver weight/body weight) × 100%.


Spleen index (%) = (spleen weight/body weight) × 100%.


Heart index (%) = (heart weight/body weight) × 100%.


### 2.12. Enzymatic Marker

After the blood collection, serum was prepared via centrifugation (5000 rpm, 5 min). LDH, CK, and CK-MB in the serum were measured using a biochemical autoanalyzer following the standard protocol for each Elisa plate.

### 2.13. Histological Assessment

Hematoxylin and eosin (HE) staining was conducted according to previous paper [54]. Tissues were fixed in 10% formalin solution over 24 h, and then embedded in paraffin. After cutting into 4 μm slices, the tissues were stained with hematoxylin solution for 3–5 min and then stained with eosin solution for 5 min. After being dehydrated, these slices were observed using an optical microscope (Eclipse E100, Nikon, Tokyo, Japan).

### 2.14. Statistical Analysis

All the experiments were conducted at least in triplicate (>3 independent experiments). The data are presented as the mean values ± standard deviation. Comparisons between groups were performed by one-way analysis of variance (ANOVA) (SPSS 19.0, IBM, Armonk, New York, USA), and *p* < 0.05 indicated statistical significance.

## 3. Result and Discussion

### 3.1. Property Assessment of PGA as a Nanocarrier

Polymers possess a large number of isomers due to the chemical complexity, which should be studied in detail to illustrate structure–property relationships [55]. For use as nanocarriers, linear PGA was synthesized from glutamic acid into two isoforms (α-PGA and γ-PGA, Figure 1a). Generally speaking, structure would influence the assembly behavior. To evaluate their entrapment ability, several different drugs, including IBU, OXC, DTX, RES, POD, NIF, MTX, and DOX, were utilized to construct drug-loaded NPs. After preparation, different results were found for α-PGA and γ-PGA. These drugs were all entrapped by α-PGA successfully, and a homogenous drug-loaded nanoparticle solution was achieved; however, when γ-PGA was used as the nanocarrier, all the samples presented precipitation (Figure 1b). Although these two PGA types possessed the same components, their different structures resulted in different physicochemical properties. As hydrophilic nanocarriers, the hydrophilicity of PGA was due to the carboxyl group. These two isoforms of PGA presented different linker chains between the main hydrophobic chain and hydrophilic carboxyl group, which induced different hydrophilic/hydrophobic volume ratios, steric hindrance, and surface charge [56,57]. When the PGA and drugs were assembled into aggregates, the different structure of isoforms affected the self-assembly process. For γ-PGA, the carboxyl group was linked with the main chain directly, which led to a low hydrophilic/hydrophobic volume ratio and high steric hindrance, making it difficult to entrap hydrophobic drugs to form stable NPs. On the contrary, α-PGA had a high hydrophilic volume/hydrophobic volume ratio and relatively low steric hindrance, enhancing the opportunity of the carboxyl group to interact with the drugs. These results revealed that α-PGA was suitable for utilization as a nanocarrier.

### 3.2. Characterization of Drug-Loaded α-PGA NPs

The drug-loaded NPs with α-PGA as the nanocarrier were researched in detail. Although all the hydrophobic drugs could be entrapped by α-PGA successfully, these drug-loaded NPs showed different results. As the drug delivery system, NPs should show a relatively small and uniform particle size; hence, based on the results of D*_h_* and PDI, α-PGA could be utilized as a nanocarrier to deliver IBU, OXC, DTX, and DOX in this experiment. Nanocarriers can encapsulate hydrophobic drugs via molecular interactions, including hydrophobic interaction, electrostatic interaction, hydrogen bond, and Van der Waals force. Therefore, these drugs could be entrapped to form nanoparticles successfully. Among these hydrophobic drug-loaded nanoparticles, PGA/DOX NPs showed the highest DLC and EE, which were 66.2 ± 4.3% and 72.7 ± 5.3%, correspondingly (Table 1). This phenomenon could be explained by the fact that α-PGA/DOX NPs were prepared via electrostatic interaction between the negative charge of PGA and the positive charge of DOX, which could be verified by the zeta potential of these drug-loaded NPs. The zeta potentials of IBU NPs, OXC NPs, and DTX NPs were negative, and the values were similar to that of free PGA. On the contrary, the zeta potential of PGA/DOX NPs was positive because the carboxyl group in PGA interacted with the amine group in DOX via electrostatic interaction, neutralizing the negative charge. Moreover, the PGA/DOX NPs presented a positive charge due to the large amount of DOX. These results prove that PGA as an anionic polymer material can be used as a nanocarrier to deliver hydrophobic drugs with a positive charge.

α-PGA/DOX NPs presented a small hydrodynamic diameter of 110.4 nm and a PDI of 0.18 (Figure 2a) and had a spherical-like structure (Figure 2b). The mechanism of this phenomenon could be the electrostatic interaction between α-PGA and DOX, which was a relatively strong interaction that led to NPs with a more compact structure. To confirm the physical interaction between α-PGA and DOX, the microstructures of α-PGA/DOX NPs were detected using the Fourier-transform infrared (FT-IR) spectra (Figure 2c). After entrapping DOX, the IR spectrum of α-PGA/DOX NPs showed the combined characteristic peaks from DOX and α-PGA, and no significant difference was shown in comparison with a physical mixture. This phenomenon could be attributed to the fact that α-PGA/DOX NPs were prepared via physical electrostatic interaction, unobservable chemical interactions were present.

According to the public reports, in general, DOX-loaded NPs showed a DLC of approximately 10–37% [45,58,59,60,61] and a particle size of approximately 100–270 nm; besides, it was difficult to prepare DOX-loaded NPs with high DLC and small particle size. Compared with these DOX-loaded NPs, α-PGA/DOX NPs presented a significantly enhanced DLC of 66.2% and a small particle size of 110 nm, which could promote its antitumor efficacy.

### 3.3. The Stability of α-PGA/DOX NPs

The stability of α-PGA/DOX NPs including the storage stability and media stability was studied. The storage stability was assessed at 4 °C for 30 days, and the particle sizes of the α-PGA/DOX NPs were detected and are recorded in Figure 3a. During the whole storage period, no stratification, turbidity, precipitation, or other phenomena were shown. After 30 days, the particle size of α-PGA/DOX NPs changed slightly (110.4 ± 18.6 vs. 117.8 ± 21.6, *p* > 0.05), and the PDI was decreased from 0.18 ± 0.02 to 0.12 ± 0.04 as storage time was prolonged (*p* > 0.05). These results indicate that α-PGA/DOX NPs showed good storage stability.

The media stability of α-PGA/DOX NPs was investigated in 0.9% normal saline, 5% glucose solution, PBS buffer solution, and mouse plasma. The results are shown in Figure 3b. α-PGA/DOX NPs were stable in 5% glucose solution and mouse plasma, in which the particle size changed slightly (*p* > 0.05), while the particle size increased sharply after adding 0.9% normal saline and PBS buffer solution (>1000 nm). This suggested that α-PGA/DOX NPs presented good media stability in 5% glucose solution and mouse plasma, indicating that α-PGA/DOX NPs could be applied via intravenous administration with 5% glucose solution.

### 3.4. Drug Release of α-PGA/DOX NPs In Vitro

The α-PGA/DOX NPs were constructed via electrostatic interaction, which may cause the release behavior to be affected by the pH value of the release medium. Hence, the release characteristics were analyzed in PBS solutions at pH 5.5 and pH 7.4 (Figure 4). There was a significant burst release of free DOX solution, and a complete release was shown within the initial 24 h. The release rate was affected by the pH value; free DOX had a high release rate in pH 5.5 PBS solution. The release of α-PGA/DOX NPs in PBS solution showed a sustained release profile for at least 6 days, and no significant burst release was shown. The DOX release of α-PGA/DOX NPs in PBS buffer (pH 7.4) was relatively slow; the cumulative release rate reached 23.4 ± 2.5% within 24 h and 59.7 ± 4.6% within 144 h. When the release medium was changed to pH 5.5 PBS, the release rate of α-PGA/DOX NPs was enhanced significantly to 40.9 ± 4.8% within 24 h (vs. pH 7.4, *p* < 0.01) and 70.4 ± 5.5% within 144 h (vs. pH 7.4, *p* < 0.01). Comparing with the neutral condition, DOX released fast in acidic condition. The reason for this pH-sensitive release was that α-PGA protonates in an acidic environment, and its affinity with DOX was weakened, resulting in a fast release. The above results indicated that α-PGA/DOX NPs have a pH-sensitive and sustained release profile, which means they could achieve a low release in blood circulation (pH 7.4) and fast release in tumor tissue (pH 5.5). This characteristic could be used for targeted drug delivery to the tumor microenvironment.

### 3.5. Antitumor Effect In Vitro

To research the antitumor effects of α-PGA/DOX NPs, the proliferation-inhibition effects of free DOX and α-PGA/DOX NPs against 4T1 cells were investigated by the CCK-8 method, and the results are shown in Figure 5. The cell inhibition rate of α-PGA was maintained at approximately 20%, indicating that the carrier had no cytotoxicity. Free DOX and α-PGA/DOX NPs showed a good dose-dependent inhibitory effect, and α-PGA/DOX NPs showed a higher inhibitory effect at the same concentration. The IC_50_ value of free DOX and α-PGA/DOX NPs was 9.09 ± 0.92 and 1.06 ± 0.77 μg mL^−1^ separately, and the cytotoxicity of α-PGA/DOX NPs was enhanced approximately 8.5 fold after 48 h incubation. The antitumor efficacy of α-PGA/DOX NPs was promoted significantly in comparison with that of free DOX (*p* < 0.001). The mechanism leading to this phenomenon could be that NPs transported more DOX into tumor cells through intracellular endocytosis and pinocytosis, compared with the passive diffusion of free DOX drugs into cells, thus avoiding the multidrug resistance of the cells to the drugs. These results revealed that α-PGA/DOX NPs significantly increased the antitumor effect of DOX.

### 3.6. Antitumor Effect In Vivo

To further evaluate the antitumor efficacy of α-PGA/DOX NPs in vivo, the tumor volume changes, tumor inhibition rate based on the tumor weight, and apoptotic situation of the tumor tissue were recorded. The tumor volume of the α-PGA group (blank control, 1950 mm^3^) showed no significant difference compared with that in the glucose solution group (negative control, 2150 mm^3^, *p* > 0.05), indicating no tumor inhibition efficacy forα-PGA (Figure 6a). DOX injection and α-PGA/DOX NPs presented good antitumor efficacy; the tumor volume increased slowly, and a significant difference was shown (** *p* < 0.01, vs. glucose solution). Moreover, α-PGA/DOX NPs exhibited higher antitumor efficacy than free DOX injection; the tumor volume change was decreased 1.4 fold (781 vs. 1104 mm^3^), which was a significant difference (^#^ *p* < 0.05).

Furthermore, the antitumor activity was evaluated via the tumor weight, and the TIR was calculated (Figure 6b). The average tumor weight was 1.8 ± 0.5, 1.6 ± 0.3, 1.0 ± 0.3, and 0.6 ± 0.2 g for glucose solution, α-PGA, DOX injection, and α-PGA/DOX NPs; the tumor inhibition rate was 11.1%, 45.4%, and 67.4% for PGA, DOX injection, and α-PGA/DOX NPs. Compared with DOX injection, α-PGA/DOX NPs had a higher antitumor activity, and the TIR was promoted 1.5 fold (^#^ *p* < 0.05).

After sacrifice, the apoptotic situation of the tumor tissue was observed via the HE staining method (Figure 6c,d). The tumor tissues of the DOX injection group and PGA/DOX NP group both presented necrosis (marked by a black line), but the degree of apoptosis was different. The tumor apoptosis was more significant in the PGA/DOX NP group, and the area of necrosis was more than that in the DOX group.

All these results indicated that α-PGA/DOX NPs showed good antitumor efficacy; this phenomenon could be explained by the properties of α-PGA/DOX NPs. α-PGA/DOX NPs could be passively targeted into tumor tissues through the EPR effect. After entering tumor cells via endocytosis, DOX was released suddenly from the NPs owing to its pH sensitivity.

### 3.7. Systemic Toxicity Test

DOX, as the commercial chemotherapeutic agent, showed significant systemic toxicity and cardiotoxicity; hence, it was necessary to study the relative toxicity of α-PGA/DOX NPs. To estimate the systemic toxicity, body weight changes and the liver/spleen index were recorded. The body weight of the four groups showed two different tendencies (Figure 7a). The body weight of the glucose solution, α-PGA, and α-PGA/DOX NP groups increased, while the DOX injection group showed decreased body weight. Compared with the glucose solution group, no significant difference was shown in the α-PGA/DOX NP group, but the body weight of the DOX group presented significant difference (*** *p* < 0.001). The α-PGA/DOX NP group also showed a significant difference compared with the DOX injection group (^##^ *p* < 0.01). These results proved that α-PGA/DOX NPs could decrease the side effects of DOX.

Then, the liver and spleen indexes were calculated to evaluate systemic toxicity furthermore, and the results are shown in Figure 7b. Compared with glucose solution, the liver index of the other three groups showed no change, suggesting that DOX, α-PGA, and α-PGA/DOX had no hepatotoxicity. The spleen index was significantly decreased for the DOX group, while it was maintained for the α-PGA and α-PGA/DOX groups, suggesting that DOX showed spleen toxicity that was not carried over in α-PGA/DOX NPs. Based on these results, α-PGA/DOX NPs could decrease systemic toxicity effectively.

### 3.8. Cardiotoxicity

To research the cardiotoxicity of α-PGA/DOX NPs, the heart weight and the heart index were measured (Figure 8a). The heart weight was 0.11 ± 0.02, 0.12 ± 0.02, 0.08 ± 0.02, and 0.13 ± 0.01 g for glucose solution group, PGA group, DOX injection group, and α-PGA/DOX NP group separately. Compared with the glucose solution group, the heart weight of the PGA group and the α-PGA/DOX NP group was slightly increased; on the contrary, the heart weight of the DOX injection group was decreased. Moreover, a significant difference was shown between the DOX injection group and the α-PGA/DOX NP group (^#^ *p* < 0.05). These results indicated that DOX caused severe heart injury and that α-PGA/DOX NPs could decrease the cardiotoxicity of DOX. Then, the relative heart index was calculated, which was 0.55 ± 0.01%, 0.56 ± 0.02%, 0.44 ± 0.03%, and 0.60 ± 0.05% for the glucose solution group, PGA group, DOX injection group, and α-PGA/DOX NP group, correspondingly. Similar to the result for heart weight, the heart index of the PGA group and the α-PGA/DOX NP group was increased and there existed no significant difference (*p* > 0.05) compared with the glucose solution group, while the heart index of the DOX injection group was decreased and showed a significant difference (*** *p* < 0.001). The difference was also presented between the DOX injection group and the α-PGA/DOX NP group (^###^ *p* < 0.001), proving that the α-PGA/DOX NP group could decrease the cardiotoxicity of DOX.

Moreover, the concentration of enzymatic markers, including LDH, CK, and CK-MB in serum, was enhanced significantly after myocardial injury and was measured using an Elisa plate (Figure 8b). Compared with the glucose solution group, the concentrations of these markers in the PGA group were maintained with no significant difference (*p* > 0.05), indicating nontoxicity of PGA. The concentration of all enzymatic markers was increased in the DOX injection group significantly (*** *p* < 0.001), revealing that free DOX caused heart injury. Although the LDH of the α-PGA/DOX NP group increased from 57.5 ± 9.5 to 99.5 ± 7.5 U mL^−1^ (** *p* < 0.01, vs. glucose solution group), the CK and CK-MB were maintained, and no difference was shown (*p* > 0.05). Besides, all these markers of the α-PGA/DOX NP group decreased significantly compared with the DOX injection group (^###^ *p* < 0.001). All these results suggested that α-PGA/DOX NPs could decrease the heart toxicity of DOX obviously.

The heart tissue was observed via the HE staining method, and the images are shown in Figure 8c,d. Significant karyolysis of the cardiomyocyte was shown in the DOX injection group, resulting in a number of cardiomyocytes that were apoptotic, and edema was observed in the heart tissue. On the contrary, normal heart tissue was presented in the α-PGA/DOX NPs.

These results revealed that DOX injection induced severe heart injury due to energy metabolism change, induction of apoptosis, intracellular calcium dysregulation, and oxidative stress [62,63], while α-PGA/DOX NPs could reduce heart injury while still delivering the drug.

## 4. Conclusions

To research the structural influence of PGA, two isoforms, α-PGA and γ-PGA, were utilized as nanocarriers to encapsulate several hydrophobic drugs. It was confirmed that α-PGA could be applied to construct a nanodrug delivery system due to its appropriate structure and relatively low steric hindrance. α-PGA as an anionic polypeptide, interacted with cationic DOX via electrostatic interactions to assemble stable α-PGA/DOX NPs with high DLC and small particle diameter. α-PGA/DOX NPs presented good storage stability, media stability, and a pH-sensitive release profile. The in vitro antitumor activity of α-PGA/DOX NPs against 4T1 cells was enhanced approximately 8.5 fold compared with DOX injection. Animal experiments indicated that α-PGA/DOX NPs promoted the antitumor activity of DOX and reduced its toxicity. Compared with DOX injection, the tumor inhibition rate of α-PGA/DOX NPs increased by approximately 1.5 fold; the systemic toxicity and cardiotoxicity of DOX were decreased significantly, no body weight loss or abnormal enzymatic markers were detected, and normal heart tissue was observed. The advantage of α-PGA/DOX NPs in this study include simple preparation, high DLC, relative small particle size, and good antitumor efficacy. In summary, as a hydrophilic anionic polypeptide, α-PGA demonstrated promising as a nanocarrier to construct nanodrug delivery systems with potential applications in the clinic.

## Figures and Tables

**Figure 1 polymers-14-02242-f001:**
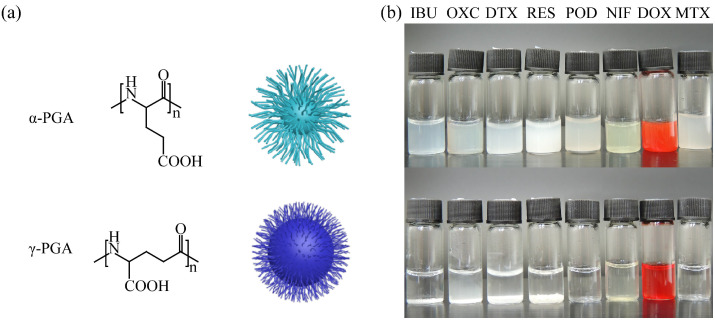
Structure of PGA and their aggregates (**a**), images of drug-loaded NPs (**b**).

**Figure 2 polymers-14-02242-f002:**
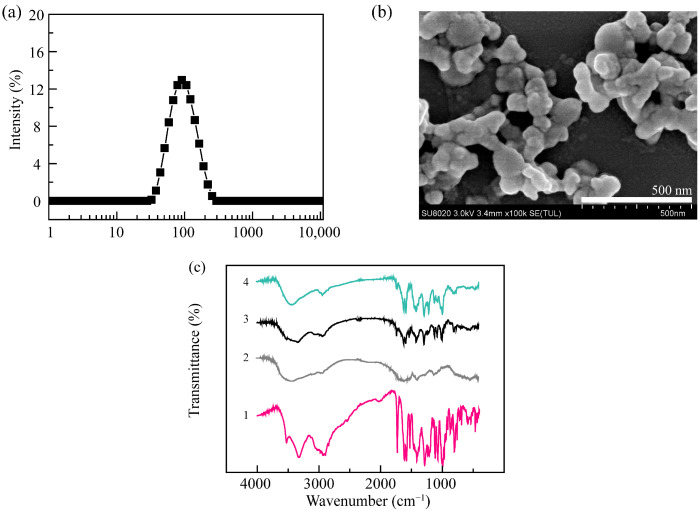
Particle size distribution curve of α-PGA/DOX NPs detected by dynamic light scattering (**a**); SEM image (**b**), scale bar 500 nm; FT-IR spectra (**c**) of DOX (1), PGA (2), physical mixture (3), and α-PGA/DOX NPs (4).

**Figure 3 polymers-14-02242-f003:**
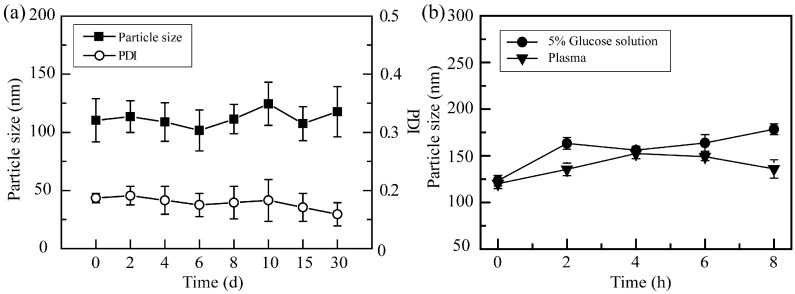
Particle size of α-PGA/DOX NPs in deionized water placed at 4 °C for 30 days (**a**); stability of α-PGA/DOX NPs in 5% glucose solution and plasma (**b**) (*n* = 3).

**Figure 4 polymers-14-02242-f004:**
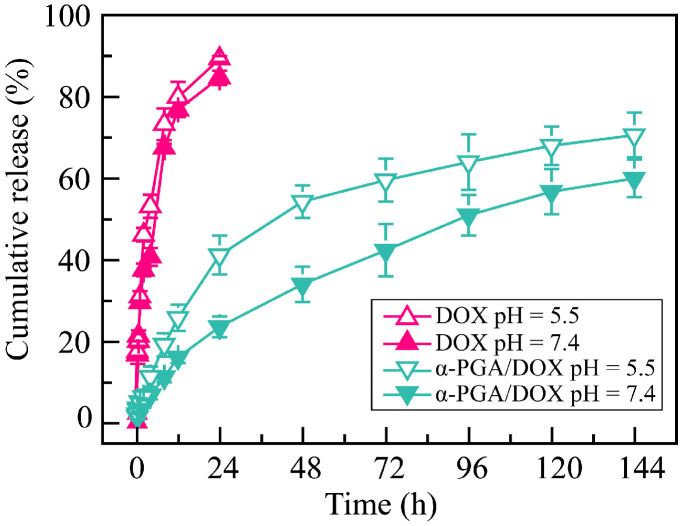
Cumulative release rate of α-PGA/DOX NPs in PBS solution (*n* = 3).

**Figure 5 polymers-14-02242-f005:**
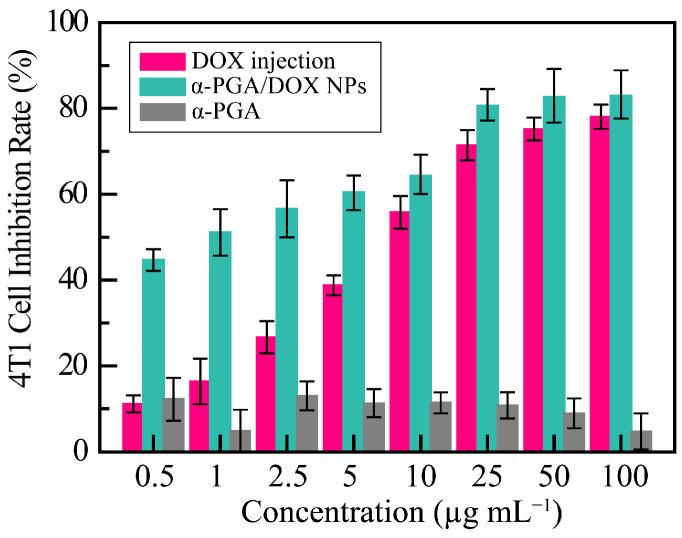
Inhibitory effect of α-PGA/DOX NPs after 48 h incubation (*n* = 5).

**Figure 6 polymers-14-02242-f006:**
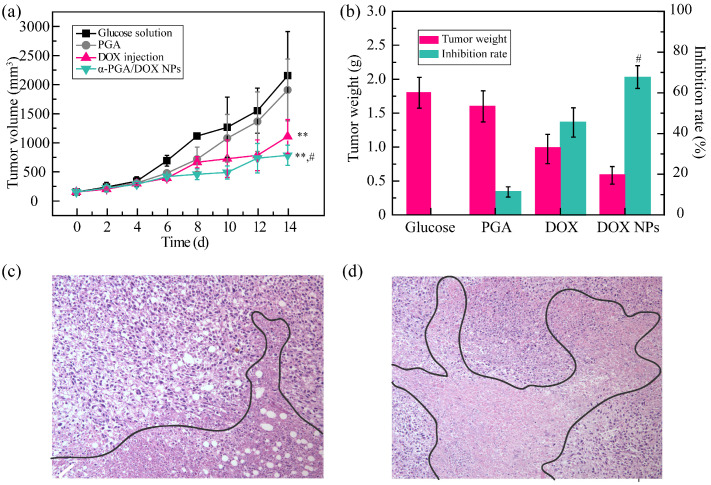
Antitumor results on 4T1 tumor-bearing mice in vivo: tumor volume change curves (**a**), tumor inhibition rate (**b**), tumor tissue images of the DOX group (**c**) and the α-PGA/DOX NP group (**d**) (*n* = 10). ** *p* < 0.01, vs. glucose solution; ^#^ *p* < 0.05, vs. DOX injection.

**Figure 7 polymers-14-02242-f007:**
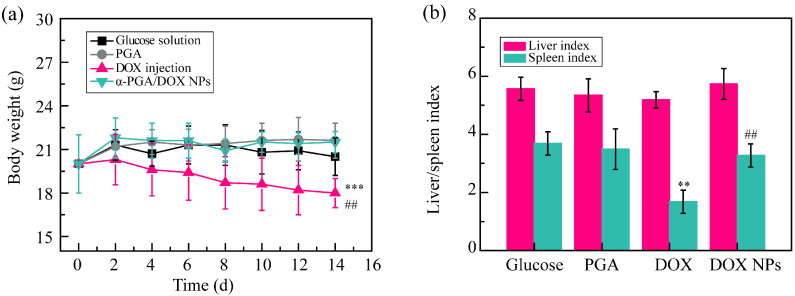
Systemic toxicity assessment: body weight change curves (**a**) and liver/spleen index (**b**) (*n* = 10). ** *p* < 0.01, *** *p* < 0.001, vs. glucose solution; ^##^ *p* < 0.01, vs. DOX injection.

**Figure 8 polymers-14-02242-f008:**
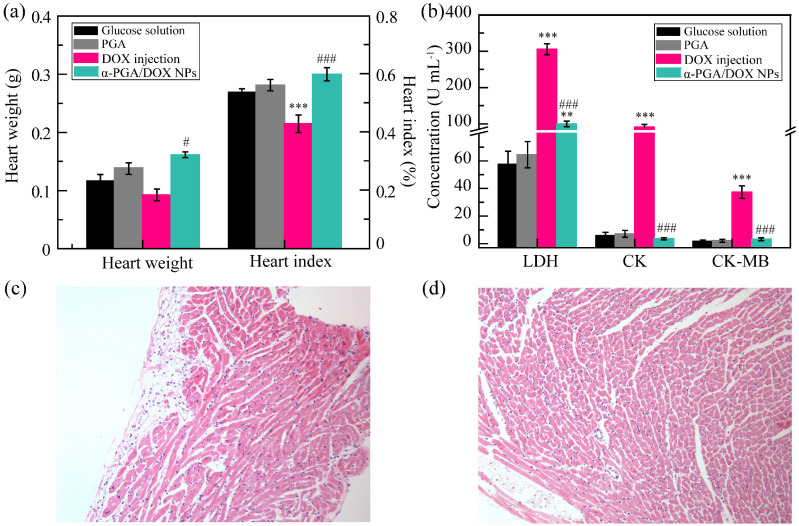
Cardiotoxicity measurement: heart weight and heart index (**a**), enzymatic marker concentration (**b**), heart tissue images of the DOX group (**c**) and α-PGA/DOX NP group (**d**) (*n* = 10). ** *p* < 0.01, *** *p* < 0.001, vs. glucose solution; ^#^ *p* < 0.05, ^###^ *p* < 0.001, vs. DOX injection.

**Table 1 polymers-14-02242-t001:** Results of the drug-loaded α-PGA NPs.

Drug	IBU	OXC	DTX	RES	POD	DOX	MTX
D*_h_* (nm) ^a^	198.9 ± 5.5	184.3 ± 9.5	396.7 ± 10.3	608.1 ± 25.9	225.3 ± 15.1	110.4 ± 18.6	418.5 ± 22.1
PDI ^a^	0.16 ± 0.03	0.32 ± 0.04	0.36 ± 0.03	0.39 ± 0.02	0.62 ± 0.08	0.18 ± 0.02	0.50 ± 0.04
ζ (mV) ^a^	−32.0 ± 0.3	−20.0 ± 0.4	−30.4 ± 0.3	31.8 ± 0.4	39.5 ± 0.4	29.0 ± 0.2	30.4 ± 0.9
DLC (%) ^b^	39.4 ± 1.1	31.0 ± 1.5	42.6 ± 2.5	43.5 ± 1.4	53.0 ± 1.7	66.2 ± 4.3	35.5 ± 1.1
EE (%) ^b^	49.2 ± 1.4	38.7 ± 1.9	54.3 ± 3.2	54.4 ± 1.8	66.2 ± 2.2	72.7 ± 5.3	44.4 ± 1.3

^a^ Dynamic light scattering detected, *n* = 3. ^b^ UV-HPLC detected, *n* = 3.

## Data Availability

Not applicable.

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
