# Peer review of "Hydrophilic Poly(glutamic acid)-Based Nanodrug Delivery System: Structural Influence and Antitumor Efficacy"

_polymers, 2022, doi:10.3390/polym14112242_

Round 1

Reviewer 1 Report

Dear authors,

After revision the article is improved, however, there are still some small issues to consider:

When I ask “Q5. What does it mean OD value? And SD? Dh?”

Isn't to explain to me because I understand but, in all manuscript, the abbreviation should be explained/descript.

In Q11. In Line 270, the assay was in vivo? (added)”

“R. In line 270, the assay was estimated in vivo.” This information is not included in this line and should be.

I suggest the inclusion of the FTIR results in the manuscript.

The conclusion could be improved, it should be short, simple, and without numerical results.

Author Response

Dear reviewer,

Thanks very much for your comment. Your suggestion is very valuable and helpful for us to improve our manuscript. All of the comments have been replied point by point, and relevant manuscript have been revised.

When I ask “Q5. What does it mean OD value? And SD? Dh?”

Isn't to explain to me because I understand but, in all manuscript, the abbreviation should be explained/descript.

Sorry to misunderstand your comment. The abbreviation of OD value (page 4, line 171), SD (page 5, line 206), and Dh (page 3, line 130) have been explained when they appear the first time.

In “Q11. In Line 270, the assay was in vivo? (added)”

“R. In line 270, the assay was estimated in vivo.” This information is not included in this line and should be.

We are sorry that the line number is forgotten to renew. After being revised, line 270 was changed to line 346 (page 10), which is section “3.6 Antitumor effect in vivo”.

I suggest the inclusion of the FTIR results in the manuscript.

Followed your suggestion, FTIR results and relevant experimental procedure have added in the revised manuscript, including section 3.2 (page 7, line 268-272), figure 2c (page 8), and section 2.7 (page 3, line 141-144).

The conclusion could be improved, it should be short, simple, and without numerical results.

Thanks very much for your suggestion. The conclusion has been shorten, the numerical results are deleted.

Reviewer 2 Report

In this article, Guo et al. produced two poly(glutamic acid) drug delivery systems for cancer therapy. The nanoparticles were characterized, and their in vitro and in vivo therapeutic efficacy was assessed. This is a resubmitted manuscript in which the authors addressed most of my previous comments satisfactorily. Please consider some minor concerns:

Line 29: “poly(glutamic acid)” and “antitumor efficacy” already appear in the title. The words should be removed or replaced by other keywords to increase the article’s visibility. For example, poly(amino acid), cancer, pH-sensitive…

Line 33: Please introduce here the abbreviation of nanoparticles (NPs) and use it throughout the article.

Lines 33-34: “Drug-loaded nanoparticles are solid colloidal particles with the diameter ranging from 10 to 1000 nm, which are constructed using drugs and nanocarriers.” Nanoparticles are nanocarriers. Please rewrite the sentence.

Line 39: This sentence should be rewritten for a better understanding as nanoparticles are nanocarriers.

Section 2.7. If the authors evaluate variations of the zeta potential of the NPs, please include this in the methods and results.

Line 185: Please provide details on centrifugation time and speed.

Section 3.2. Please discuss the zeta potential values.

Section 3.2. Looking for the NPs characterization, for which drugs can α-PGA be used as DDS? Please add this discussion to the manuscript.

Author Response

Dear reviewer,

Thanks very much for your comment. Your suggestion is very valuable and helpful for us to improve our manuscript. All of the comments have been replied point by point, and relevant manuscript have been revised.

Line 29: “poly(glutamic acid)” and “antitumor efficacy” already appear in the title. The words should be removed or replaced by other keywords to increase the article’s visibility. For example, poly(amino acid), cancer, pH-sensitive…

Thanks very for your suggestion. Keywords, including poly(glutamic acid) and antitumor efficacy, have been replaced by poly(amino acid) and pH-sensitive (page 1, line 29).

Line 33: Please introduce here the abbreviation of nanoparticles (NPs) and use it throughout the article.

The abbreviation of nanoparticles (NPs) is introduced in line 34 (page 1).

Lines 33-34: “Drug-loaded nanoparticles are solid colloidal particles with the diameter ranging from 10 to 1000 nm, which are constructed using drugs and nanocarriers.” Nanoparticles are nanocarriers. Please rewrite the sentence.

The sentence is changed to “Drug-loaded nanoparticles (NPs) are constructed by hydrophobic drugs and am-phiphilic nanocarriers via molecular interactions [4,5], the particle size is ranging from 10 to 1000 nm.” (page 1, line 34-37).

Line 39: This sentence should be rewritten for a better understanding as nanoparticles are nanocarriers.

The sentence is changed to “As a component of drug delivery system, nanocarriers play an important role” (page 1, line 42).

Section 2.7. If the authors evaluate variations of the zeta potential of the NPs, please include this in the methods and results.

We are sorry for our unclear description in last response. When the storage stability is evaluated, the zeta potential of NPs is recorded, but unfortunately, the data is only measured at 0 and 30 d, which is 29.0 ± 0.2 and 31.7 ± 1.1 mV. Due to our careless, the data is not recorded during the entire storage process. When the media stability is detected, zeta potential is not recorded. We totally agree with you that the zeta potential should be measured and utilized as a parameter to evaluate the stability of NPs, which will be recorded in our future study.

Line 185: Please provide details on centrifugation time and speed.

Centrifugation time is 5 min and speed is 5000 rpm. The details have been added in page 5, line 195.

Section 3.2. Please discuss the zeta potential values.

These discussion “This phenomenon could be explained that α-PGA/DOX NPs was prepared via electrostatic interaction between negative charge of PGA and positive charge of DOX, which could be verified by zeta potential of these drug-loaded NPs. Zeta potential of IBU NPs, OXC NPs, DTX NPs were negative and the value were similar as free PGA. On the contrary, zeta potential of PGA/DOX NPs was positive because the carboxyl group in PGA was interacted with amine group in DOX via electrostatic interactions, hence, the negative charge was neutralized, PGA/DOX NPs presented positive charge due to the large amount of DOX.” have been added in section 3.2 (page 6, line 251-256).

Section 3.2. Looking for the NPs characterization, for which drugs can α-PGA be used as DDS? Please add this discussion to the manuscript.

These discussion “As the drug delivery system, NPs should show relative small and uniform particle size, hence, based on the results of Dh and PDI, α-PGA could be utilized as nanocarriers to delivery IBU, OXC, DTX, and DOX in this experiment. To load hydrophobic drug via physical entrapment, nanocarriers can encapsulate hydrophobic drugs via molecular interactions, including hydrophobic interaction, electrostatic interaction, hydrogen bond, and Van der Waals force. Therefore, these drugs could be entrapped to form nanoparticles successfully. Among these hydrophobic drug-loaded nanoparticles, PGA/DOX NPs showed the highest DLC and EE, which was 66.2 ± 4.3% and 72.7 ± 5.3% correspondingly (Table 1). This phenomenon could be explained that α-PGA/DOX NPs was prepared via electrostatic interaction between negative charge of PGA and positive charge of DOX, which could be verified by zeta potential of these drug-loaded NPs. Zeta potential of IBU NPs, OXC NPs, DTX NPs were negative and the value were similar as free PGA. On the contrary, zeta potential of PGA/DOX NPs was positive be-cause the carboxyl group in PGA was interacted with amine group in DOX via electro-static interaction, inducing the negative charge was neutralized, PGA/DOX NPs pre-sented positive charge due to the large amount of DOX. These results prove that PGA as an anionic polymer material can be used as nanocarriers to delivery hydrophobic drugs with positive charge.” have been added in the section 3.2 (page 6, line 237-257).

This manuscript is a resubmission of an earlier submission. The following is a list of the peer review reports and author responses from that submission.

Round 1

Reviewer 1 Report

The manuscript addresses a relevant but also rather competitive topic. Globally, the work is well structured and presented. However, it raises some critical concerns in relation to the scientific novelty or lack thereof. The authors managed to point out the relevance of the subject, but do not provide any convincing rationale for studying a system that is well-known. Indeed, neither the proposed polymeric nanocarrier nor the target polymer-drug conjugate are new. Poly(glutamic acid), PGA, in both of its forms (γ-PGA and α-LPGA), conjugated with doxorubicin (DOX) are widely reported as polymer-drug systems in the treatment of cancer. Therefore, the substantial criticism rests on the marginal/incremental novelty that can be claimed. No new data are presented on the polymer synthesis or the formulation of the nanoparticles. The putative ``structural influence'' of the departure polymer (γ-PGA versus α-LPGA, sec. 3.1) is discussed in an essentially qualitative way. There are no data proving the electrostatic interaction that authors invoke as the mechanism underlying of the conjugate formation. Why is chemical conjugation excluded? The work is nothing but an extension, with some archival value, of similar or even more complete studies involving PGA-DOX-based systems. This is far from being a striking novelty to justify its publication in Polymer. In view of this, the reviewer considers that the manuscript does not have scientific content and should be rejected. 

Reviewer 2 Report

Dear Authors,

The manuscript entitled "Hydrophilic poly(glutamic acid)-based nanodrug delivery system: structural influence and antitumor efficacy” is an interesting topic to improve tumor therapy.

I suggest some important changes to better understand the article.

Q1. What is the EPR effect? (Line 35)

I suppose it is “enhanced permeability and retention”, but it is important to clarify…

It is also important to define various abbreviations, such as: DLC, OEG, DMF, PDI, PBS (in line 64, 65, 100, 117, and 129 respectively).

Q2. When they address the method of preparation of the nanoparticles they do not mention the environment... the temperature at which they were produced...

I suggest you improve the description of the method.

Q3. Improve the materials section. There are materials mentioned where it is not known where they were acquired.

Q4. The methods of dialysis and anti-tumor effect followed protocols. The protocols used were either developed by your team or adapted from protocols already reviewed in the literature. You should refer to the authors of the protocols.

Q5. What does it mean OD value? (Line 148) And SD? (Line 170)

Q6. If abbreviations have already been identified they should be used and not repeated. (Line 170)

Q7. The legend should be improved and separated into (a) structure… (b) image (Line 197)

Q8. Line 200 - Avoid starting a sentence with “then”

Q9. Table 1

- should be presented after mentioned and not before.

- What does Dh mean?

Q10. How was the image (a) in figure 2 obtained?

Q11. Antitumor or anti-tumor? (Line 254, 270)

In Line 270, the assay was in vivo? (added)

Q12. Why didn't they do an MTT or MTS assay to assess cell viability?

Q13. Add histological analysis in the methods section.

Explain HE meaning… (Line 287)

Q14. How was the number of animals used estimated (n =10)?

How were the animals sacrificed?

The methods section should be improved, as should the discussion.

The Fourier-transform infrared spectroscopy characterization should have been carried out.

The discussion should be improved by comparing the results obtained with other studies, where other nanocarriers are used.

In the conclusion, the advantage of using these nanocarriers compared to others should be explained.

Reviewer 3 Report

In this article, Guo et al. produced two types of poly(glutamic acid) drug delivery systems for cancer therapy. The nanoparticles were characterized, and their in vitro and in vivo therapeutic efficacy was assessed. Although the topic is interesting, I have a few concerns about some of the methodologies used and the presented data. Numerous issues need to be addressed before publication.

First, thorough editing of this article needs to be done since there are several stylistic, grammatical errors, and typos.

- Section 2.3. The authors should provide more details. What is the final drug concentration? Why was the dialysis bag used? Separate something? Please clarify.

- The authors should determine the encapsulation efficiency of all drug-loaded α-PGA NPs. Add the methodology associated with it.

- Section 2.5. The authors should provide more details.

- Section 2.7. Did the authors evaluate variations of the PDI and zeta potential of the NPs?

- Section 3.1. The section does not have any references. References should be added to support the discussion.

- Table 1. Zeta potential values are missing. Please present all data as mean ± SD.

- Section 3.2. Looking for the characterization of the NPs (including encapsulation efficiency), for which drugs can α-PGA be used as DDS?

- Please make clear the scale in Figure 2b.

- Sections 3.3, 3.4, and 3.5 are discussed without statistical analysis. The authors should perform it and discuss the result according to the p-values (all p-values should be included in the manuscript).

- Figure 3 corresponds to which medium? Please clarify in this section and the Method.

- Section 2.8. “α-PGA/DOX NPs aqueous solution and DOX solution”. Which medium and which DOX concentration? According to the results, the release at pH 5.5 was also evaluated.

- Line 225: “α-PGA/DOX NPs were stable in 5% glucose solution”. Figure 3b shows a considerable increase (50 nm).

- Figure 5: Control group with glucose solution is missing and should be added. Without it, the authors can not conclude that “the carrier had no cytotoxicity”.

- Figure 8: The group of PGA is missing. Please add and discuss the data.

- Line 334: “no difference was shown between glucose solution group and α-PGA/DOX NPs group”. P-value? The increase seems significant.

- Lines 336-338: This experiment is not in the methodology. Please add it.

- Lines 339-340: !enhanced slightly in α-PGA/DOX NPs”. The CK concentration is lower than in the glucose group.

- Lines 341-342: “no significant difference was shown in α-PGA/DOX NPs (p > 0.05, vs. glucose solution)”. Nor for the LDH marker?

- Line 348: “All these results revealed DOX injection induced the severe heart injury”. Any explanation for that?

Minor revisions:

Lines 14-17: This sentence should be rewritten for a better understanding.

Line 218: “poly(glutamic acid)” already appears in the title. The word should be removed or replaced by other keywords to increase the article’s visibility. For example, poly(amino acid), cancer, pH-sensitive…

Lines 32-33: “Drug-loaded nanoparticles are solid colloidal particles (…) which are constructed by drugs and polymer materials”. Polymers are not the only type of material used for nanoparticles production. Metallic and lipid nanoparticles are other examples. Please rewrite the sentence.

Line 35: The abbreviation “EPR” was not introduced.

Line 38-39: This sentence should be rewritten for a better understanding.

Line 46: Please use “biocompatibility and biodegradability” instead of “biocompatibility, biodegradability”

Line 48: Please use “which can interact” instead of “which can be interacted”.

Line 56: “α-PGA is synthesized by chemical synthesis”. The sentence is redundant. Please rewrite it.

Line 64: The abbreviation “DLC” was not introduced. Introduce it in line 63.

Line 66: “ were” instead of “can be”.

Line 101: The abbreviation “DMF” was not introduced.

Line 116: “These drug-loaded nanoparticles were diluted”. Which medium?

Line 117: The abbreviation “PDI” was not introduced.

Line 121: The abbreviation “NPs” was not introduced before. Please standardize the manuscript.

Line 127: Please use “days” instead of “d” throughout the article.

Line 127: “were recorded at 0, 2, 4, 6, 8, 10, 12, and 14 d”. Figure 3 shows measurement after 30 days.

Line 164: How were animals sacrificed?

Line 17: Please use “Property assessment” instead of “assessment property”.

Lines 177-178: Drug abbreviations were already introduced before.

Line 204: The abbreviation “DLC” was already introduced.

Line 270: Please standardize “Antitumor” or “anti-tumor” throughout the article.

Lines 292 and 332: Please use “showed” or similar instead of “emerged”.

Line 315: Please use “no hepatotoxicity” instead of “low or no hepatotoxicity”.

Lines 327-331: Please rewrite these sentences for a better understanding.